# Stability Study of mRNA-Lipid Nanoparticles Exposed to Various Conditions Based on the Evaluation between Physicochemical Properties and Their Relation with Protein Expression Ability

**DOI:** 10.3390/pharmaceutics14112357

**Published:** 2022-10-31

**Authors:** Mariko Kamiya, Makoto Matsumoto, Kazuma Yamashita, Tatsunori Izumi, Maho Kawaguchi, Shusaku Mizukami, Masako Tsurumaru, Hidefumi Mukai, Shigeru Kawakami

**Affiliations:** 1Graduate School of Biomedical Sciences, Nagasaki University, 1-7-1 Sakamoto, Nagasaki 852-8588, Japan; 2Under Graduate School of Pharmaceutical Sciences, Nagasaki University, 1-14 Bunkyo, Nagasaki 852-8521, Japan; 3Department of Immune Regulation, SHionogi Global INfectious DiseasEs Division (SHINE), Institute of Tropical Medicine, Nagasaki University, 1-12-4 Sakamoto, Nagasaki 852-8523, Japan; 4Department of Hospital Pharmacy, Nagasaki University Hospital, 1-7-1 Sakamoto, Nagasaki 852-8501, Japan

**Keywords:** mRNA-LNP, storage condition, stability

## Abstract

Lipid nanoparticles (LNPs) are currently in the spotlight as delivery systems for mRNA therapeutics and have been used in the Pfizer/BioNTech and Moderna COVID-19 vaccines. mRNA-LNP formulations have been indicated to require strict control, including maintenance at fairly low temperatures during their transport and storage. Since it is a new pharmaceutical modality, there is a lack of information on the systematic investigation of how storage and handling conditions affect the physicochemical properties of mRNA-LNPs and their protein expression ability. In this study, using the mRNA-LNPs with standard composition, we evaluated the effects of temperature, cryoprotectants, vibration, light exposure, and syringe aspiration from the vials on the physicochemical properties of nanoparticles in relation to their in vitro/in vivo protein expression ability. Among these factors, storage at −80 °C without a cryoprotectant caused a decrease in protein expression, which may be attributed to particle aggregation. Exposure to vibration and light also caused similar changes under certain conditions. Exposure to these factors can occur during laboratory and hospital handling. It is essential to have sufficient knowledge of the stability of mRNA-LNPs in terms of their physical properties and protein expression ability at an early stage to ensure reproducible research and development and medical care.

## 1. Introduction

In mRNA therapeutics, the genetic information of a therapeutic protein is administered in the form of mRNA, which is then translated into a protein in the cytoplasm to produce a pharmacological effect [1,2]. This offers many advantages in the development of mRNA therapeutics. First, compared to conventional gene therapy, there are no concerns regarding insertion into the host genome [3,4]. Second, because there is no need to transfer genetic information into the nucleus, therapeutic proteins can be efficiently expressed, even in cells with low mitotic potential. Third, mRNA therapeutics is the mRNA itself, which encodes therapeutic proteins, and can be designed and synthesized in a short period [5,6]. The COVID-19 vaccine took less than a year from when the coronavirus genome sequence was reported to the first UK approval following clinical trials [7]. Encouraged by the success of the COVID-19 vaccine, mRNA therapeutics are expected to be applied to various vaccines and to treat other diseases, such as cancer and allergies, in the future [8,9].

The first in vivo protein expression using mRNA dates back to 1990 when Wolff et al. reported the expression of chloramphenicol acetyltransferase and luciferase in murine muscle by direct administration of naked mRNA [10]. However, it is difficult to express sufficient amounts of protein to achieve a therapeutic effect with the administration of naked mRNA. This is because mRNA is a large molecule of approximately 10^5^–10^6^ Da and has a high density of negative charge, making it difficult for mRNA to pass through the cell membrane [11]. Furthermore, mRNA is an unstable molecule that is easily degraded; therefore, its formulation is essential for the development of mRNA therapeutics. Lipid nanoparticles (LNPs) are currently in the spotlight as a formulation technology and have been used in Pfizer/BioNTech and Moderna’s COVID-19 vaccine. They are nanoparticles composed mainly of ionized lipids, helper lipids, cholesterol, and PEG-lipids, with encapsulated mRNA [3,12,13]. LNPs provide efficient cellular uptake by endocytosis and release of internalized mRNA into the cytoplasm by promoting escape from the endosomes and protecting mRNA from degradation by RNase.

mRNA-LNP formulations have been indicated to require strict controls, including the need to maintain fairly low temperatures during transport and storage [12,13]. It has been reported that when mRNAs alone are stored at room temperature, there is a slight decrease in translation efficiency in vitro, even after 14 days under conditions in which RNase is strictly removed [14]. In contrast, it has been reported that external physical stimuli may cause nanoparticle aggregation and mRNA release from nanoparticles, which significantly reduces the translation efficiency [15]. Since it is a new pharmaceutical modality, there is a lack of information on the systematic investigation of how storage conditions affect the physicochemical properties of mRNA-LNPs and their protein expression ability.

In this study, we selected a lipid formulation with ionized lipid DLin-MC3-DMA as our standard LNP. The lipid composition as ionized lipid: DSPC: Cholesterol: DMG-PEG2000 = 50:10:38.5:1.5 mol% is developed siRNA-LNP formulation for Onpattro by Alnylam Pharmaceuticals [16]. This composition has been described as a benchmark for nucleic acid-encapsulated LNPs [17] and is widely referenced, including Moderna’s vaccine. We selected various physical conditions that may affect storage stability throughout the entire process, from production in a factory to inoculation, referring to those listed by Kis Z [18], and exposed mRNA-LNP to these conditions to investigate their effects on physicochemical properties and protein expression ability. Specifically, the effects of storage temperature, cryoprotectant, vibration, light exposure, and syringe aspiration from vials on nanoparticle size polydispersity index, zeta potential, mRNA retention, and in vitro and in vivo protein expression using luciferase as a reporter were evaluated.

## 2. Materials and Methods

### 2.1. Cell Culture

HepG2 cells were purchased from the RIKEN BioResource Center (Ibaraki, Japan). The cells were grown in DMEM supplemented with 10% *v*/*v* FBS and 1000 units/mL penicillin, and 100 µg/mL streptomycin. The cells were incubated in a humidified atmosphere of 5.0% CO_2_ at 37 °C.

### 2.2. Mice

Male wild-type ddY mice aged 5 weeks were purchased from SLC (Shizuoka, Japan). The animal experiments were approved by the Guidelines for Animal Experimentation of Nagasaki University and the Institutional Animal Care and Use Committee of Nagasaki University (approval number: 1812251497-7).

### 2.3. mRNA In Vitro Transcription

mRNA was produced using standard in vitro transcription, according to previously established methods [19] and the manufacturer’s recommendations. Briefly, plasmid DNA encoding firefly luciferase with poly-A tail was digested by restriction enzyme Sap I, followed by the production of mRNA using the HiScribe T7 High Yield RNA Synthesis Kit (New England Biolabs, Ipswich, MA, USA) with CleanCap^®^ Reagent AG (TriLink BioTechnologies, NorthPark, CA, USA), according to the manufacturer’s instructions. Thereafter, mRNA was purified.

### 2.4. Preparation of mRNA Encapsulating LNP

mRNA-LNPs were prepared as previously described [20,21] with some modifications. The aqueous phase was prepared using 50 mM citrate buffer (pH 3.0) and the luciferase mRNA. The ethanol phase was prepared with four lipid components: ionizable lipid DLin-MC3-DMA (MedChemExpress, Monmouth Junction, NJ, USA), 1,2-distearoyl-sn-glycero-3-phosphocholine (DSPC, Avanti, Alabaster, AL, USA), cholesterol (NACALAI TESQUE, Kyoto, Japan), and DMG-PEG2000 at a molar ratio of 50:10:38.5:1.5. The aqueous and ethanol phases were mixed using a NanoAssemblr Benchtop (Precision NanoSystems Inc., Vancouver, BC, Canada) with a microfluidic device at a 1:3 volume ratio. The mRNA-LNP was transferred immediately to SnakeSkin™ Dialysis Tubing (10,000 MWCO, Thermo Fischer Scientific Inc., Waltham, MA, USA) and dialyzed overnight at 4 °C in PBS (pH7.4) to remove ethanol. The volume of PBS buffer was approximately 500–1000 × the sample fraction volume. After dialysis, the sample was concentrated using Amicon Ultra (100,000 Da, Merck KGaA, Darmstadt, Germany) and filtered through a 0.45 μm membrane filter.

### 2.5. Measurement of LNP Characterization

The particle size was measured using the intensity-averaged particle size (Z-average), polydispersity index (PDI), and zeta potential on a ZetaSizer Nano (Malvern Panalytical, Malvern, UK). The percentages of encapsulated mRNA and mRNA concentrations were determined using the Quant-iT RiboGreen RNA Reagent Kit (Life Technologies, Waltham, MA, USA), as previously described [19]. To quantify the amount of RNA inside the LNP, the formulation was diluted 100-fold in TE buffer and added 10% *v*/*v* Polyoxyethylene(10) octylphenyl ether (Wako Pure Chemicals Industries, Osaka, Japan). The manufacturer’s instructions were used to quantify RNA. For quantification of encapsulation efficiency, RNA concentrations obtained from samples without Polyoxyethylene(10) octylphenyl ether were interpreted as “not encapsulated” because the dye cannot breach lipid membranes, and fluorescence from Polyoxyethylene(10) octylphenyl ether-treated samples represented the total RNA amount (outside and inside LNPs). Fluorescence (excitation at 485 nm and emission at 530 nm) was detected using a SpectraMax iD3 microplate reader (Molecular Devices, San Jose, CA, USA).

### 2.6. In Vitro Luciferase Assay

Briefly, HepG2 cells were seeded in a 48-well cell culture plate at 4 × 10^3^ cells/cm^2^, and after 24 h, mRNA-LNPs were added as mRNA 0.1 μg/well. After 48 h post-seeding in the plate, cells were washed with cold PBS twice and lysed with 150 μL lysis buffer for 20 min on ice, centrifuged, and corrected 100 μL supernatant. Luciferase activity was measured using PicaGene (Toyobo, Osaka, Japan) and normalized to protein concentration.

### 2.7. mRNA Extraction from mRNA-LNPs and Agarose Gel Electrophoresis

To disrupt the lipid nanoparticles, 100 µL of mRNA-LNPs were added to 4 µL of 10% *v*/*v* Polyoxyethylene(10) octylphenyl ether (Wako Pure Chemicals Industries) as a comparable product of Triton X-100, vortexed briefly, and incubated for 15 min. Lithium chloride (8 M) was added to half the volume of mRNA-LNP and incubated for an hour at room temperature. The solution was then centrifuged at 20,600× *g* for 30 min at 4 °C. The supernatant was discarded, and the pellet was rinsed with 70% ethanol twice. The pellet was dried in vacuo and resuspended in 20 µL of RNase-free water. 0.2 µg mRNA was added 10 µL loading buffer consisting of 50% formamide, 0.4 M formaldehyde, 1 × MOPS running buffer, and 0.02% bromophenol blue.

### 2.8. In Vivo Imaging Luciferase Activity

The mice were shaved to reduce photon scattering and absorption by fur. Fifty microliters of mRNA-LNPs were injected intramuscularly. Mice were intraperitoneally injected with 150 μL D-luciferin solution (30 mg/mL, Syd Labs, Boston, MA, USA) in PBS, and in vivo bioluminescent imaging was performed at 10 min of D-luciferin administration using an exposure time of a minute with an IVIS Lumina II (Caliper Life Sciences, Hopkinton, MA, USA) for three consecutive days. The total flux in the region of interest (ROI) was determined using the Living Image 3.0 software (Caliper Life Sciences).

### 2.9. In Vivo Luciferase Assay

The luciferase assay was performed as previously described [22,23]. Briefly, 2 μg of mRNA-LNPs in PBS was intramuscularly injected. After 4.5 h, the mice were sacrificed, and the liver and muscle treated with mRNA-LNP were collected. The tissues were washed twice with cold PBS and homogenized in lysis buffer. Homogenates were centrifuged, and luciferase activity was measured using PicaGene (Toyobo) and normalized to the protein concentration.

### 2.10. Statistical Analysis

Statistical analyses were performed using one-way ANOVA with Tukey’s or Dunnett’s test. The data are shown as mean ± standard deviation (S.D.). * *p* < 0.05 and ** *p* < 0.01 were considered statistically significant and extremely statistically significant. All in vitro data are representative of at least three independent experiments.

## 3. Results

### 3.1. Time-Dependent Physical Stability of mRNA-LNP

First, we evaluated the time-dependent physical properties of mRNA-LNPs. The hydrodynamic diameter of the LNP was 112.2 nm, the zeta potential of the LNP was −5.723 mV, and the encapsulation efficacy was 93.96% (Table 1). During the seven days, the hydrodynamic diameter, zeta potential, and encapsulation efficacy of the mRNA-LNPs were retained. 

### 3.2. Temperature Affected the Stability of LNPs

We evaluated their stability under aqueous conditions at several temperatures. Pfizer’s COVID-19 mRNA vaccine is recommended to be stored at −90 to −60 °C, and Moderna’s COVID-19 mRNA vaccine is recommended to be stored at −25 to −15 °C [12,13]. Therefore, the LNPs were stored either in a deep freezer (−80 °C), freezer (−30 °C), refrigerator (4 °C), or at room temperature (25 °C) for 7 days. After 7 days, the stock solutions were brought to the refrigerator for 4 h, then evaluated for their physical properties. 

The LNPs stored at −80 °C resulted in an increase in the Z-average and PDI and a decrease in encapsulation efficacy (Table 2) and luciferase expression (Figure 1). However, the LNPs stored at 4 and 25 °C maintained the Z-average, PDI, and EE, decreasing in luciferase expression.

### 3.3. Cryoprotectants Increased the Ultra-Raw Storage Stability of LNPs

In the interest of cryoprotectants, Pfizer/BioNTech and Moderna vaccines with sucrose to a final concentration of 10% (*w*/*v*) [12,13] and nanoparticles with sucrose did not affect their physicochemical properties and siRNA abilities [24]. Further experiments were performed by adding sucrose to prevent aggregation of the LNPs stored at −80 °C. The addition of sucrose improved the physical properties (Table 3) and luciferase expression of LNPs stored at −80 °C (Figure 2).

### 3.4. Vibration Affected the Stability of mRNA-LNPs

Another key factor in mRNA-LNP storage is vibration. Pfizer/BioNTech and Moderna vaccines must not be shaken [12,13]. Thus, we verified the effect of weak-to-strong vibrations on mRNA-LNPs. We assumed tapping to be a weak vibration and a vortex to be a strong vibration. The strong vibration also increased the particle size and PDI and leaked encapsulated mRNA compared to the weak vibration for the same 5 min (Table 4). The weak and short strong vibrations did not affect luciferase expression (Figure 3).

### 3.5. Light Exposure Decreased the Stability of mRNA-LNPs

Pfizer/BioNTech and Moderna vaccines have been reported to be unstable in light. However, no detailed information on the effects of light exposure on mRNA-LNPs has been published. Therefore, we investigated the properties of mRNA-LNPs and luciferase expression under closely specified conditions. Under 25 °C conditions, 20 h of exposure under normal fluorescent light (1000 lx, total 2 × 10^4^ lx·h) was compared with 20 h of exposure at 6000 lx (total 1.2 × 10^6^ lx·h) to allow the reaction to proceed under very specific conditions. Light exposure did not change particle size, PDI, or mRNA inclusion rate (Table 5); however, luciferase expression tended to decrease in a light exposure-dependent manner (Figure 4).

### 3.6. The Suction Pressure from Vials Did Not Affect the Stability of mRNA-LNPs

Salmin et al. reported that mRNA-LNPs aspirated in syringes maintained their physicochemical properties for about a day, independent of syringe material, but the strong mechanical stress of manually pumping 40 times resulted in particle aggregation [25]. We prepared our experiment to determine whether LNPs could aggregate due to aspiration pressure, as they were collected from the vial through a syringe needle. We collected 50 µL each from the same vial using five syringes and a syringe needle and investigated the physicochemical properties and biological activity of the particles. The results showed no significant changes in particle size, PDI, mRNA inclusion rate (Table 6), or luciferase expression in the HepG2 cells (Figure 5).

### 3.7. mRNA Degradation into mRNA-LNPs

Considering that the decreased expression of mRNA-LNP is due to the degradation of the primary structure of mRNA, agarose gel electrophoresis was performed to confirm the integrity of the mRNA strand length [26,27]. Among the storage conditions that decreased expression in cells with mRNA-LNP, the −80 °C storage group, which may have resulted in LNP aggregation, and the light-exposed group, which showed a possible decrease in mRNA activity, were compared.

The mRNA strand length of interest was 1950 nt, and bands of the same height were identified in the untreated, light-exposed, and −80 °C storage groups. These results indicated that the primary structure of the mRNA was not degraded (Figure 6).

### 3.8. In Vivo Luciferase Activity

Several reports show that mRNA-LNPs accumulate at the intramuscular administration site and in the liver [28,29,30]. Firefly luciferase-encoded mRNA-loaded LNP were administered intramuscularly into mice on the right leg. Luciferase-mediated bioluminescence emission was imaged at 0.5, 4.5, 9, 24, and 48 h post-injection (Figure 7a), and the total flux in the region of interest (ROI) in the abdominal region (Figure 7b) and right leg was used as the administration site (Figure 7c). The total flux indicated a reduction in luciferase activity of mRNA-LNPs exposed to light and stored at −80 °C in both the abdominal region and right leg. 

Thereafter, the liver and muscles administrated with mRNA-LNPs were removed, and their luciferase activities were quantitatively evaluated. Firefly luciferase-encoded mRNA-LNPs were administered intramuscularly to mice on the right leg, and 4.5 h later, the liver and right leg muscles were removed. These tissues were homogenized with lysis buffer and centrifuged, the supernatant was collected, and luciferase activity was measured. Similar to the in vitro results, mRNA-LNPs exposed to light and stored at −80 °C showed a reduction in luciferase activity in both the liver (Figure 7d) and muscle (Figure 7e).

## 4. Discussion

In this study, using the mRNA-LNPs with standard composition, we evaluated the effects of temperature, cryoprotectants, vibration, light exposure, and syringe aspiration from the vials on the physicochemical properties of nanoparticles in relation to their in vitro/in vivo protein expression ability. We were able to clearly identify alterations in the physicochemical properties and protein expression of the mRNA-LNPs by setting up an unusual situation, such as freezing without a cryoprotectant. For instance, storage at −80 °C without a cryoprotectant caused a decrease in protein expression that may be attributed to particle aggregation exposure to vibration and similar changes under certain conditions. It was suggested that for factors capable of aggregating mRNA-LNPs, care should be taken during research, development, and operation in the clinic.

Regarding the effects of storage temperature, the effects of storage for 7 days and subsequent thawing on the physicochemical properties and protein expression ability of the mRNA-LNPs were examined, mainly by assuming storage in the laboratory during research and development and after delivery to hospitals. Storage at −80 °C without cryoprotectant caused aggregation of mRNA-LNPs (Table 2), and luciferase expression in HepG2 cells in vitro was markedly reduced (Figure 2). Similarly, after intramuscular administration of mRNA-LNPs stored under these conditions in mice, luciferase expression was extremely low compared to that of freshly prepared mRNA-LNPs, and only slight expression was observed in the muscle at the site of administration. However, mRNAs that were extracted from mRNA-LNPs after storage at −80 °C were shown to be intact by agarose gel electrophoresis. This suggests that disruption of the LNP structure may be the cause of reduced protein expression, which is in good agreement with a previous report that mRNA-LNP aggregation leads to decreased expression in cells [10]. During freezing, the concentration of nanoparticles increases because of ice formation, and the distance between particles is reduced [31]. Furthermore, the ionic strength increases, hindering the electrostatic repulsive force in the interaction between the particles. These may contribute to aggregation [32]. Indeed, the addition of sucrose as a cryoprotectant inhibited aggregation and reduced the protein expression ability of mRNA-LNPs after storage at −80 °C (Table 3). Since sucrose is also added to Pfizer/BioNTech and Moderna’s COVID-19 vaccines, storage at −80 °C with a cryoprotectant may be effective for continuous experimentation with the same lot of mRNA-LNPs during research and development in the laboratory. 

In contrast, storage at 4 and 25 °C did not change the amount of mRNA retained in the LNPs for 7 days, nor did the physicochemical properties change (Table 1). However, in vitro luciferase expression activity was significantly reduced in a temperature-dependent manner (Figure 1). This decrease in mRNA-LNP quality may be explained by previously reported oxidation of lipids and mRNAs. As it has been reported that the oxidation of lipids constituting LNPs also induces oxidation of mRNAs enclosed inside them, LNPs using easily oxidizable lipids may induce more oxidation of mRNA [3]. It has been reported that mRNA activity is significantly decreased by mRNA oxidation and hydrolysis due to alterations in mRNA structure [33,34]. Furthermore, alterations in the mRNA structure lead to a reduction in protein expression [35]. For these reasons, it is recommended that mRNA-LNPs be properly cryopreserved (with the addition of cryoprotectant at −80 °C) when stored for more than a few days, even in laboratory research and development. 

The vibration of mRNA-LNPs in solution was studied by assuming operations prior to administration in the clinic and in the laboratory during research and development. Brief tapping, which is within the range recommended by the Pfizer/BioNTech and Moderna vaccines, did not affect either the physical properties or in vitro protein expression ability of mRNA-LNPs. However, for vortex mixing, the mRNA-LNPs used in this study tolerated exposure for 30 s; however, after 5 min of continuous exposure, a clear collapse of the nanoparticle structure was observed. Vortex mixing is a common operation in the laboratory; however, it should be avoided or kept as brief as possible. Selmin et al. reported that strong mechanical stress, such as vortexing, leads to LNP aggregation and degradation of the enclosed mRNA [25], which is consistent with the results of this study. In addition, the possibility of aggregation was examined by collecting from a vial; however, the physicochemical properties and biological activity of LNP did not change each time when collected from the same vial on five separate occasions (Figure 5). It was confirmed that this pressure level did not lead to a decrease in the activity of the particles.

With regard to light exposure, no clear changes appeared in mRNA-LNP particle size and its uniformity, zeta potential, strand length of the mRNA, and retention in the LNPs, even after 20 h of exposure to light levels six times higher than those in normal laboratory or hospital environments (Figure 7). However, in vitro expression of mRNA-LNPs decreased in a light exposure dose-dependent manner (Figure 4), and mRNA-LNPs after 1.2 × 10^6^ lx·h irradiation were only slightly expressed at the site of administration after intramuscular administration to mice. It has been reported that light irradiation induces oxidation of lipids and mRNAs, and it is possible that qualitative changes in mRNA-LNPs due to oxidation are responsible for the reduced protein expression ability [3,35], similar to that after storage at 25 or 4 °C for 7 days. It should be considered that the ionized lipid DLin-MC3-DMA used in this study contains two double bonds and is susceptible to oxidation [16]. On the other hand, ALC-0315 and SM-102, the ionized lipids currently used by the Pfizer/BioNTech and Moderna vaccines, do not have unsaturated bonds in their structure [12,13] and are expected to be less susceptible to oxidation. Actually, the Pfizer/BioNTech vaccines are indicated as being able to be refrigerated for up to one month after thawing [12]. We clearly demonstrated how oxidation affects mRNA-LNPs ability by using ionized lipids sensitive to oxidation. However, mRNA-LNPs using ionized lipids with structures more resistant to oxidation are expected to be less affected by storage at 4 and 25 °C and exposure to light, which are thought to cause oxidation. Even so, storage under light-shielding conditions is recommended for both the Pfizer/BioNTech and Moderna COVID-19 vaccines. Light exposure to mRNA-LNPs should be minimized even in the laboratory by frequent light shielding. 

The composition of the mRNA-LNPs used in this study did not exactly match that of Pfizer/BioNTech and Moderna’s COVID-19 vaccines; there were differences in various additives. In addition, ionizable lipids with various chemical properties are currently being used in mRNA-LNP formulations under research and development; therefore, the results of this study cannot be generalized to all mRNA-LNP formulations. However, the composition of the mRNA-LNPs used in this study is currently the standard, and the findings obtained regarding their stability when exposed to various environments are expected to be somewhat similar to those of other mRNA-LNP formulations. Furthermore, the results of this study suggest that the decrease in mRNA-LNP quality due to lipid or mRNA oxidation, which is not easy to detect, can occur during laboratory and hospital handling. It would be possible to realize highly reproducible research and development and medical treatment by evaluating the physicochemical properties and protein expression ability after storage in advance in the development process of mRNA-LNP formulations, as in the present study.

## 5. Conclusions

In this study, we showed that storage of mRNA-LNPs with a standard lipid composition at −80 °C without cryoprotectant caused a decrease in protein expression, possibly due to nanoparticle aggregation, and that exposure to vibration and light also caused similar changes under some conditions. Exposure to these factors may occur during laboratory and hospital handling. It is essential to have sufficient knowledge of the stability of mRNA-LNPs in terms of their physical properties and protein expression ability at an early stage to ensure reproducible research, development, and medical care. Therefore, this information is valuable for the clinical use of mRNA-LNPs.

## Figures and Tables

**Figure 1 pharmaceutics-14-02357-f001:**
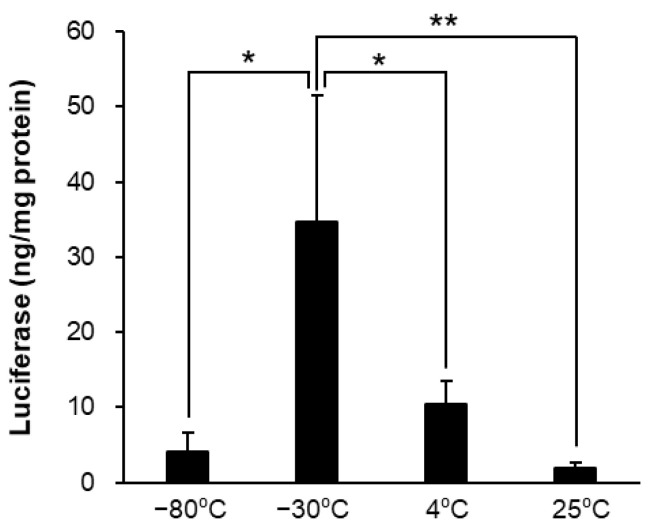
Effects of the storage temperature for in vitro luciferase assay. The mRNA-LNP storage was preserved at −80, −30, 4, and 25 °C for 7 days and thawed. Firefly luciferase expression in HepG2 cells was measured post 24 h adding of each mRNA-LNPs. Data are represented as mean + S.D. (*n* = 3). * *p* < 0.05, ** *p* < 0.01 indicate significant differences (Tukey’s test).

**Figure 2 pharmaceutics-14-02357-f002:**
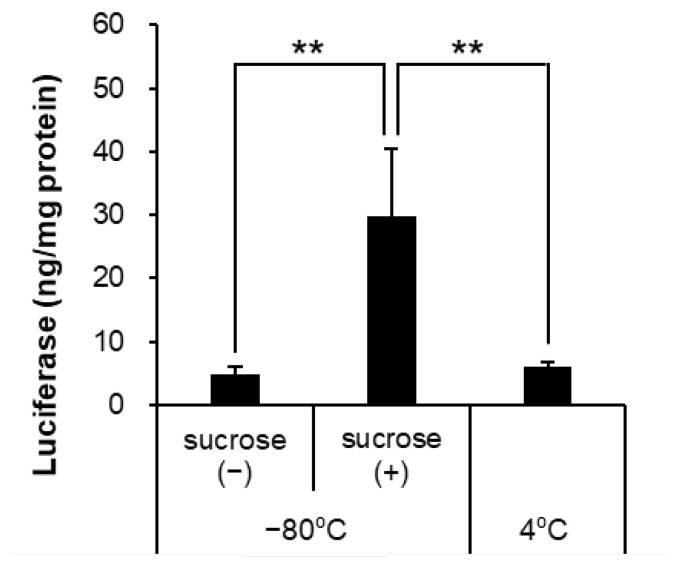
Effects of the cryoprotectants against freeze-thaw at −80 °C for in vitro luciferase assay. The mRNA-LNP storage was preserved at −80 °C (added sucrose or not) or 4 °C for 7 days and thawed. Firefly luciferase expression in HepG2 cells was measured post-24 h, adding each mRNA-LNPs. Data are represented as mean + S.D. (*n* = 3). ** *p* < 0.01 indicate significant differences (Tukey’s test).

**Figure 3 pharmaceutics-14-02357-f003:**
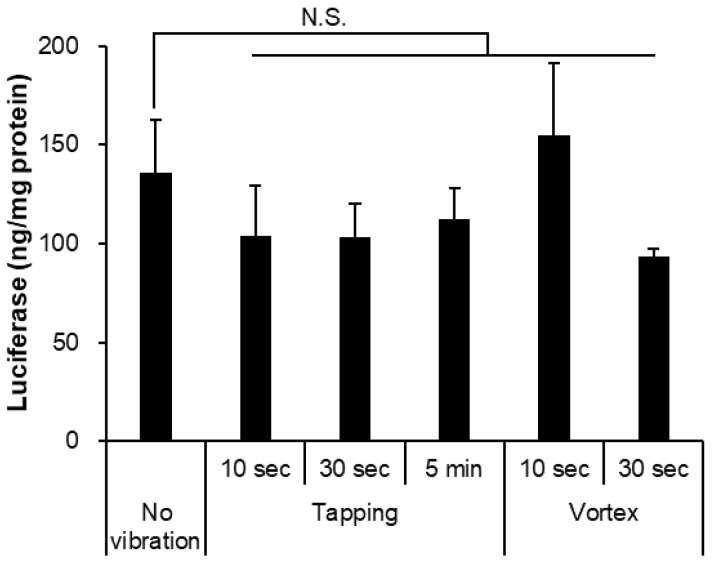
Effects of the vibration for in vitro luciferase assay. The mRNA-LNP treated tapping or vortex with 10 s, 30 s, and 5 min, respectively. Firefly luciferase expression in HepG2 cells was measured at post 24 h adding of each mRNA-LNPs. The LNP-treated vortex for 5 min leaked mRNA substantially and was not able to remain for assay. Data are represented as mean + SD (*n* = 3), N.S. indicates not significant versus the non-treatment (Dunnett’s test).

**Figure 4 pharmaceutics-14-02357-f004:**
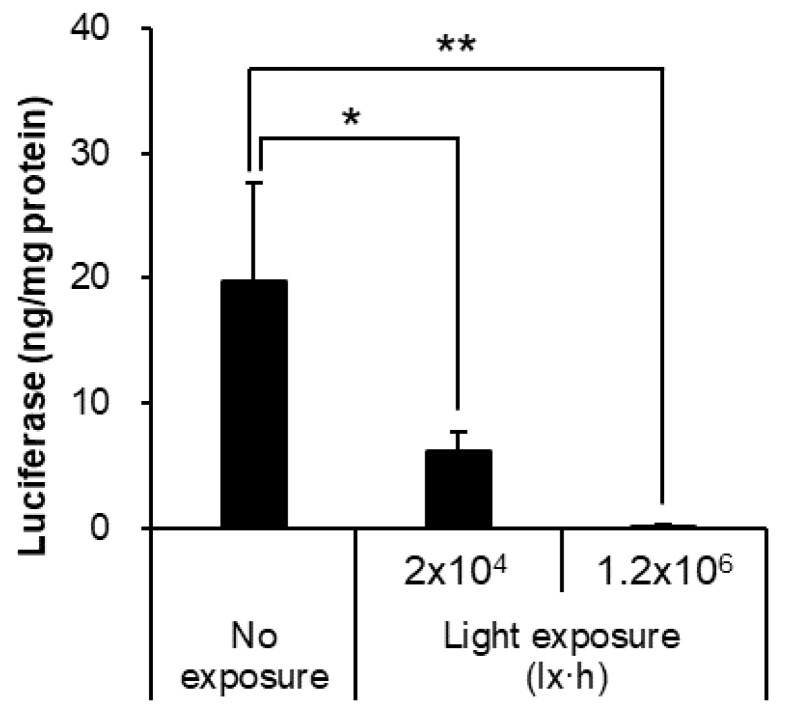
Effects of light exposure for in vitro luciferase assay. The mRNA-LNP exposed to light (2 × 10^4^ lx·h, 1.2 × 10^6^ lx·h) or not, respectively. Firefly luciferase expression in HepG2 cells was measured at post 24 h adding of each mRNA-LNPs. Data are represented as mean + S.D. (*n* = 3). ** *p* < 0.01, * *p* < 0.05 indicate significant differences versus no exposure (Dunnett’s test).

**Figure 5 pharmaceutics-14-02357-f005:**
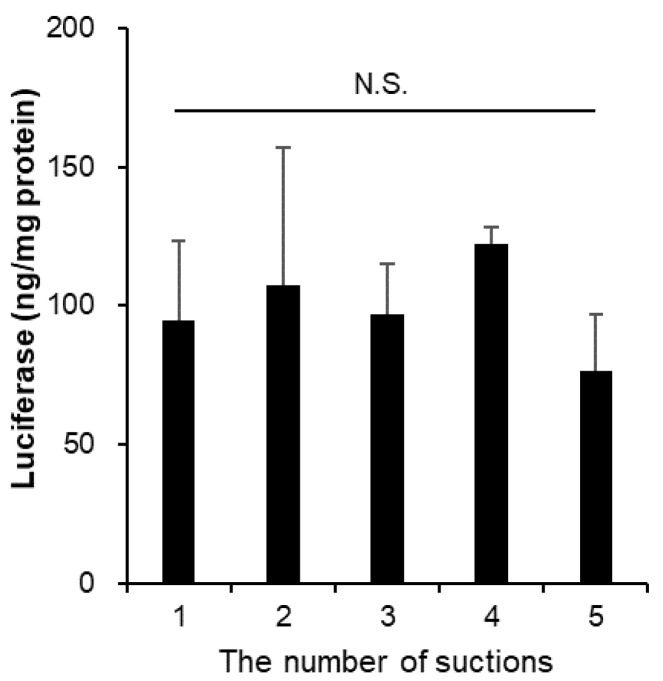
Effects of the suction pressure from the vials for in vitro luciferase assay. The mRNA-LNPs were suctioned five times from a vial through the 25 G syringe needles. Firefly luciferase expression in HepG2 cells was measured at post 24 h adding of each mRNA-LNPs. Data are represented as mean + S.D. (*n* = 3). N.S. indicates not significant (Tukey’s test).

**Figure 6 pharmaceutics-14-02357-f006:**
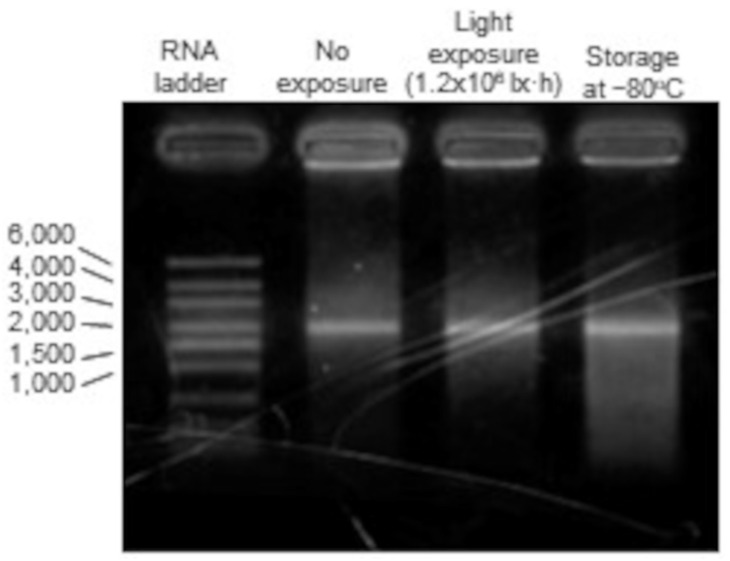
Agarose gel electrophoresis for mRNA inside LNP. Polyoxyethylene(10) octylphenyl ether was added to mRNA-LNPs in the no exposure, the light exposure (1.2 × 106 lx·h), and the storage at −80 °C groups to disrupt the membrane structure of the LNPs. mRNA strand lengths enclosed in the LNPs were confirmed by agarose gel electrophoresis in MOPS buffer.

**Figure 7 pharmaceutics-14-02357-f007:**
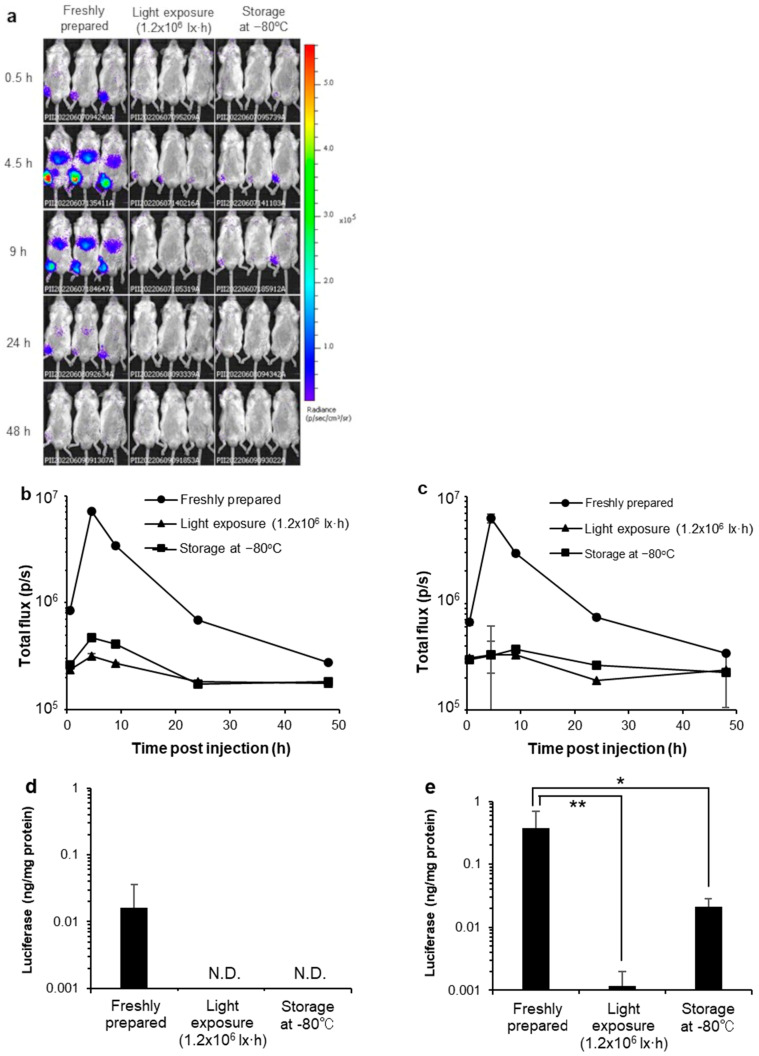
Effects of storage condition for the expression kinetics of firefly luciferase encoded mRNA-LNP and in vivo luciferase assay. (**a**) the duration and translational pattern of mRNA-LNPs were freshly prepared, exposed to the light of 1.2 × 10^6^ lx·h, and stored at −80 °C without cryoprotectants. The total flux in the region of interest (ROI) at (**b**) the abdominal region and (**c**) their right leg as the administration site. Mice were administrated intramuscularly for mRNA-LNPs, which were freshly prepared, exposed to the light of 1.2 × 10^6^ lx·h, and stored at −80 °C. After 4.5 h, their tissues were corrected, homogenized, and measured their luciferase activities with (**d**) the liver and (**e**) the muscles. The expression of the liver pattern of mRNA-LNPs, which were non-treated (control), exposed to the light of 1.2 × 10^6^ lx·h, and stored at −80 °C. Data are represented as mean + SD (*n* = 5). ** *p* < 0.01, * *p* < 0.05 indicate significant differences versus non-treatment (Dunnett’s test). N.D. indicate non-detected.

**Table 1 pharmaceutics-14-02357-t001:** The time-dependent physical stabilities of mRNA encapsulating LNPs.

Time (Days)	Size (nm)	PDI	Zeta-Potential (mV)	mRNA Retention Rate (%)
0	112.2 ± 1.3	0.081 ± 0.022	−5.72 ± 1.41	93.96 ± 1.19
1	114.9 ± 2.9	0.086 ± 0.039	−6.29 ± 2.34	95.67 ± 5.06
3	113.1 ± 2.2	0.061 ± 0.018	−6.13 ± 1.26	93.00 ± 2.50
5	111.6 ± 2.0	0.047 ± 0.043	−4.88 ± 0.67	95.45 ± 0.96
7	112.5 ± 1.0	0.065 ± 0.027	−4.15 ± 1.62	95.58 ± 1.46

Data are represented as the mean ± S.D., *n* = 3. Statistical analyses were performed using Tukey’s test.

**Table 2 pharmaceutics-14-02357-t002:** Effects of the storage temperature on particle properties of the mRNA encapsulating LNPs.

Storage Temperature	Size (nm)	PDI	Zeta-Potential (mV)	mRNA Retention Rate (%)
−80 °C	904.6 ± 107.7 **	0.695 ± 0.055	−20.06 ± 1.01	50.51 ± 10.37 **
−30 °C	170.8 ± 6.8	0.235 ± 0.028	−4.20 ± 1.19	92.97 ± 2.48
4 °C	112.5 ± 2.3	0.081 ± 0.040	−5.46 ± 1.43	97.34 ± 0.73
25 °C	113.4 ± 6.8	0.083 ± 0.005	−6.61 ± 2.15	96.72 ± 0.40

Data are represented as the mean ± S.D., *n* = 3. ** *p* < 0.01 indicate significant differences versus 4 °C (Dunnett’s test).

**Table 3 pharmaceutics-14-02357-t003:** Effects of the cryoprotectants on particle properties of the mRNA encapsulating LNPs.

Storage Temperature and Cryoprotectants	Size (nm)	PDI	Zeta-Potential (mV)	mRNA Retention Rate (%)
−80 °C, Sucrose (−)	383.2 ± 41.9 **	0.593 ± 0.065	−10.65 ± 0.92	50.83 ± 11.24 **
−80 °C, Sucrose (+)	118.4 ± 5.9	0.098 ± 0.011	−8.71 ± 0.60	83.66 ± 3.22
4 °C	110.7 ± 1.5	0.065 ± 0.038	−7.75 ± 2.75	87.25 ± 4.30

Data are represented as the mean ± S.D., *n* = 3. ** *p* < 0.01 indicate significant differences versus 4 °C (Dunnett’s test).

**Table 4 pharmaceutics-14-02357-t004:** Effects of the vibration on particle properties of the mRNA encapsulating LNPs.

Vibration	Size (nm)	PDI	Zeta-Potential (mV)	mRNA Retention Rate (%)
Non-treatment	114.5 ± 2.9	0.076 ± 0.027	−10.69 ± 5.29	89.84 ± 1.24
Tapping 10 s	120.5 ± 20.7	0.139 ± 0.081	−8.73 ± 3.19	93.51 ± 3.09
Tapping 30 s	114.7 ± 4.5	0.084 ± 0.055	−8.93 ± 1.86	91.85 ± 2.13
Tapping 5 min	119.6 ± 5.2	0.154 ± 0.067	−11.58 ± 4.51	85.62 ± 4.04
Vortex 10 s	115.5 ± 6.4	0.100 ± 0.043	−9.29 ± 2.21	88.29 ± 2.59
Vortex 30 s	116.7 ± 5.9	0.085 ± 0.024	−11.47 ± 1.01	93.69 ± 1.39
Vortex 5 min	242.8 ± 55.3 **	0.427 ± 0.202	−15.35 ± 0.17	41.58 ± 30.61 **

Data are represented as the mean ± S.D., *n* = 3. ** *p* < 0.01 indicate significant differences versus the non-treatment (Dunnett’s test).

**Table 5 pharmaceutics-14-02357-t005:** Effects of the light exposure on particle properties of the mRNA encapsulating LNPs.

Condition	Size (nm)	PDI	Zeta-Potential (mV)	mRNA Retention Rate (%)
No exposure	116.8 ± 5.8	0.045 ± 0.016	−4.37 ± 0.32	93.06 ± 2.32
Light exposure (2 × 10^4^ lx·h)	116.1 ± 6.8	0.079 ± 0.060	−4.71 ± 2.29	94.77 ± 1.54
Light exposure (1.2 × 10^6^ lx·h)	114.3 ± 5.1	0.050 ± 0.009	−5.24 ± 1.06	95.11 ± 1.30

Data are represented as the mean ± S.D., *n* = 3. Statistical analysis was performed using Dunnett’s test versus the no-exposure group.

**Table 6 pharmaceutics-14-02357-t006:** Effects of the suction from the vials on particle properties of the mRNA encapsulating LNPs.

The Number of Suctions	Size (nm)	PDI	Zeta-Potential (mV)	mRNA Retention Rate (%)
1	90.6 ± 2.2	0.072 ± 0.020	−11.16 ± 0.37	91.47 ± 1.57
2	90.5 ± 1.5	0.062 ± 0.040	−10.81 ± 1.55	89.52 ± 4.22
3	92.0 ± 1.8	0.074 ± 0.016	−12.08 ± 1.95	94.44 ± 2.09
4	91.4 ± 1.1	0.071 ± 0.026	−10.18 ± 2.13	92.29 ± 2.88
5	90.2 ± 1.8	0.064 ± 0.016	−11.53 ± 1.80	92.81 ± 2.57

Data are represented as the mean ± S.D., *n* = 3. Statistical analyses were performed using Tukey’s test.

## Data Availability

Not applicable.

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
