# Peer review of "Stability Study of mRNA-Lipid Nanoparticles Exposed to Various Conditions Based on the Evaluation between Physicochemical Properties and Their Relation with Protein Expression Ability"

_pharmaceutics, 2022, doi:10.3390/pharmaceutics14112357_

Round 1

Reviewer 1 Report

This article deals with a subject that is highly relevant for health care professionals as it points out the importance of properly storing and usig mRNA-LNP drug products. Moderna and Pfizer-BioNTech have published little data on the background of the handling directions of the mRNA-LNP as published in the inserts.

I miss a reference to the Selmin paper which was published before the Kudsiova paper. https://doi.org/10.3390/pharmaceutics13071029a

The choice of DLin-MC3-DMA is unfortunate as it is not used in the Moderna/Pfizer-BioNTech vaccines. DLin-MC3-DMA contains unsaturated acyl chains and is much more sensitive to oxidation than the ionizable lipids in the marketed products. This makes the outcome of these studies difficult to extrapolate to the products that are actually used in practice. I invite the authors to point this out/discuss the impact of their choice. As an example: figure 2: the decrease observed after 1 week storage at 4 C is significant. However, the marketed products have (much) longer approved storage times at 4C. FDA Comirnaty: ‘Thawed vials can be stored in the refrigerator [2ºC to 8ºC (35ºF to 46ºF)] for up to 10 weeks prior to use’.

The use of formulations without sucrose is not logical. It is generally known that a cryoprotectant is needed when freezing LNP formulations. Moreover, PBS is not a first choice anyway because of significant drops in pH during the freezing process in the absence of sucrose. I invite the authors to clarify in the text why they have chosen these undesirable conditions.

The numbers in the tables contain too many digits that are statistically not-significant and should be removed.

The authors may have a look at lines 223-224.

Figure 3 legend, line 234-235.

326-330… rephrase.

Author Response

We are grateful to Reviewer 1 for the critical comments and useful suggestions that have helped us to improve our paper. As indicated in the responses that follow, we have taken all these comments and suggestions into account in the revised version of our paper.

Reviewer 2 Report

In this article, the authors studied the effects of temperature, cryoprotectants, vibration, light exposure, and syringe aspiration on the physicochemical properties of lipid nanoparticles and evaluated the protein expression ability under different conditions. It was a systematic investigation and provided useful information. Some questions:  

1.     line 103-107, the authors prepared four lipid components. What are the selection criteria of these four? 

2.     I think the products of LNPs really need certain cryoprotectants when storing in low temperatures. So, is it better to discuss the effect of different cryoprotectants on the stability of mRNA-LNPs rather than to compare with or without cryoprotectants? 

Author Response

We are grateful to Reviewer 2 for the critical comments and useful suggestions that have helped us to improve our paper. As indicated in the responses that follow, we have taken all these comments and suggestions into account in the revised version of our paper.
